# The Spatial Relationship Between Two Sympatric Pheasant Species and Various Human Disturbance Activities

**DOI:** 10.3390/ani15010095

**Published:** 2025-01-03

**Authors:** Lanrong Wang, Yuting Lu, Yinfan Cai, Liling Ji, Dapeng Pang, Meisheng Zhou, Yang Cheng, Faguang Pu, Baowei Zhang

**Affiliations:** 1School of Life Sciences, Anhui University, Hefei 230601, China; d22201015@stu.ahu.edu.cn (L.W.); kexingluovo@163.com (Y.L.); yinfancai@gmail.com (Y.C.); lling_ji@163.com (L.J.); pia@163.com (D.P.); 2Anhui Tianma National Nature Reserve, Lu’an 237300, China; ttzzms@163.com (M.Z.); chengyang1101@163.com (Y.C.); margerymarcos3068@outlook.com (F.P.)

**Keywords:** anthropogenic disturbances, sympatric distribution, activity rhythm, coexistence mechanisms

## Abstract

In this study, we used camera traps to investigate the effects of different human disturbance activities on the spatiotemporal relationships between sympatric Reeves’s pheasant and Koklass pheasant in the Anhui Tianma National Nature Reserve, China. We found that the existing human disturbance activities in the reserve altered the behavioral patterns of both pheasant species and intensified interspecific competition but had no significant impact on their spatial or temporal distribution patterns.

## 1. Introduction

Globally, many areas with well-preserved environments and wildlife habitats have been designated as protected areas to maintain ecosystem biodiversity and ecological balance [1,2]. The protection of these areas can safeguard endangered species and critical habitats from human activity. However, residents living near protected areas cannot be prevented from entering these areas. Factors such as free-ranging livestock [3,4,5], domestic dogs [6,7], and occasional human harvesting [8] in mountain areas can also affect wildlife within protected areas. Livestock farming exacerbates habitat loss and degradation [3,9,10] and alters the daily activity patterns and behaviors of wildlife [11,12]. In addition, livestock compete with wild animals for limited food and space, threatening their survival. Domestic dogs further disrupt local wildlife communities through predation, competition, harassment, and the spread of diseases [13,14]. Understanding the interactions between various human disturbances and wildlife is crucial for improving the management of protected areas and maintaining wildlife populations and their habitats [15,16]. This knowledge will aid in the development of effective conservation strategies to ensure the long-term sustainability of the species and ecosystems within these areas.

Reeves’s pheasant (*Syrmaticus reevesii*) and Koklass pheasant (*Pucrasia macrolopha*) are large terrestrial forest birds that are under national key protection and are important components of the ecosystem [17]. Their population dynamics not only play a crucial role in maintaining ecosystem stability but also reflect the health of the local ecological environment [18]. Additionally, these two species have limited flying ability, making it difficult for them to form differentiated utilization patterns in vertical space. Thus, their ecological niches are narrow, and they are sensitive to human disturbances, which may lead to intense interspecies competition [19,20,21]. Therefore, they are ideal groups for studying the coexistence of human activities and wildlife. Previous studies focused primarily on the impact of livestock on large ground-dwelling bird species. For example, in the Yubaiting Nature Reserve, studies revealed a positive correlation between the presence of green peafowl and a moderate number of cattle and goats [10]. Similarly, in the Wanglang National Nature Reserve, livestock did not significantly impact blood pheasants, suggesting that livestock do not directly affect the habitat use of pheasants [12]. Overall, studies suggest that pheasant species likely maintain neutral or positive associations with livestock grazing [22]. However, research on the effects of other forms of disturbance, such as human activities and domestic dogs, remains limited. Further analysis is required to better understand these effects.

To understand the impact of various human disturbances on wildlife, this study analyzed 19 months of camera trap data from the Anhui Tianma National Nature Reserve, China, focusing on two large ground-dwelling forest birds: the Reeves’s pheasant and Koklass pheasant. By examining data from both temporal and spatial perspectives, we assessed how human disturbances have affected these two pheasant species and explored their responses to different types of human activity. The aim of this study was to provide scientific evidence to improve conservation strategies. Two key questions were addressed: how these two co-occurring pheasant species coexist through temporal and spatial niche partitioning and how they respond to the various disturbances caused by humans, livestock, and dogs across different time periods and spatial scales.

## 2. Materials and Methods

### 2.1. Study Area

This study was conducted in the Anhui Tianma National Nature Reserve in Jinzhai County, Anhui province, China (31°10′–31°20′ N, 115°20′–115°50′ E, hereinafter referred to as “TM”). TM is situated in the heart of the Dabie Mountains at the junction of the Anhui, Hubei, and Henan provinces, covering an area of 289.2 km^2^. The terrain is rugged, with the highest peak reaching an elevation of 1729.3 m above sea level. This region has a humid continental monsoon climate typical of East China, with an average annual temperature of 12.5 °C and average annual precipitation of 1832.8 mm [23]. The area is home to a diverse range of wildlife, including the Reeves’s pheasant and Koklass pheasant, as well as notable species such as the Anhui musk deer (*Moschus anhuiensis*), raccoon dog (*Nyctereutes procyonoides*), badger (*Arctonyx collaris*), masked palm civet (*Paguma larvata*), yellow weasel (*Mustela sibirica*), wild boar (*Sus scrofa*), and Reeve’s Muntjac (*Muntiacus reevesi*) [23]. The general elevation range evaluated in this study spans 400–1400 m and encompasses broadleaf, mixed coniferous and broadleaf, and coniferous forests.

The study area is protected, and hunting, mining, and illegal collection of forest resources are strictly prohibited. However, the indigenous people in the study area still rely on grazing and gathering within nature reserves for their livelihoods. Consequently, livestock grazing and collection activities have become the most common forms of human disturbance in this region.

### 2.2. Data Collection

Infrared camera trap systems (Ereagle E3H, Shenzhen, China) were used to detect wildlife and human disturbances from May 2022 to December 2023. We established a sampling array consisting of 280 grid units (1 × 1 km) and planned to install 1–2 camera stations in each unit. Considering the terrain and wildlife activity frequency in the TM, it was challenging to place the cameras precisely at the grid center. Therefore, researchers carefully selected the best camera installation sites along ridges or valleys based on their proximity to animal trails and the presence of animal droppings. A minimum linear distance of 500 m was maintained between infrared cameras. Ultimately, 270 infrared camera stations were used to monitor the reserves (Figure 1).

The cameras were fixed on tree trunks at a height of 30–50 cm, facing north and parallel to the ground. The cameras were set for 24 h continuous monitoring at medium sensitivity; once triggered, they would take three consecutive photos and record a 10 s video, with a 30 s interval before the next photo [24].

### 2.3. Environmental Covariates

Based on previous research [4,10,12,25], we selected the following covariates to evaluate their influence on occupancy: elevation, slope, enhanced vegetation index (EVI), distance to the nearest settlement, and distance to the nearest road (Table 1). These covariates encompass both environmental and anthropogenic factors and comprehensively reflect the habitat characteristics and quality. (1) Elevation and slope directly influence the terrain and climatic conditions of the habitat [26]. (2) EVI indicates vegetation cover and density, which can be used to understand the ecological characteristics of the habitat. (3) Roads and human settlements are widespread disturbances in the study area and may impact habitat quality [26]. These variables were extracted from a digital elevation model of the reserve (https://www.gscloud.cn/ accessed on 24 September 2023; resolution: 30 × 30 m) and the National Catalogue Service for Geographic Information of China (https://www.webmap.cn/ accessed on 7 October 2023). The EVI was obtained from Landsat 8 satellite imagery through Google Earth Engine, with the average value from May to August used for analysis.

### 2.4. Data Analysis

#### 2.4.1. Single-Season, Single-Species Data Analysis

Infrared cameras and occupancy models are effective tools for analyzing the relationship between species distribution and habitat characteristics, particularly spatial interactions between species, niche differentiation, and the multiple factors influencing species distribution [27,28]. We used single-species occupancy models to analyze the habitat selection preferences of two pheasant species.

To construct a single-species, single-season occupancy model [29,30,31], we used infrared camera monitoring data collected from May to August 2022 and May to August 2023. The study period was limited to May to September for two main reasons: first, to ensure that the timeframe met the model’s single-season closure requirement [32], and second, because this period falls between summer and autumn, when food is abundant and the climate is warm, facilitating the analysis of interspecies competition [33,34]. Third, it covers the breeding season of pheasants, when animal activity is frequent, allowing for better capture of habitat use patterns and territorial behavior [35]. Occupancy modeling was constructed using the “unmarked” package in R (version 4.4.1) software (R Foundation, Vienna, Austria) [36,37].

Historical records were established for the target species with 20 detections every 8 days. For any images captured during each segment, 1 indicated “detected”, 0 indicated “not detected”, and NA indicated a camera trap malfunction [38]. Considering that insufficient monitoring time may significantly affect model fitting, we excluded sites with fewer than six detections (fewer than 48 camera days) [34,39], resulting in a dataset of 251 sites for the analysis.

We checked for collinearity among the covariates using Spearman’s rho and selected those with a correlation coefficient of |r| < 0.7 for further analysis [34,40]. The model results were ranked using Akaike’s Information Criterion (AIC), and models with ΔAIC ≤ 2 were considered the best models for evaluating species habitat preferences [41]. The final results were based on the weighted averages of all the best models.

#### 2.4.2. Two-Species Occupancy Modeling

A two-species occupancy model was established to analyze the effects of human disturbance on the occupancy of Reeves’s Pheasant and Koklass Pheasant without considering the detection probability [42]. First, based on the best combination of single-species and single-season occupancy models, a two-species single-season occupancy model with eight parameters was established (Table A1) [43]. According to model assumptions and relevant studies, livestock, humans, and dogs were treated as dominant species (species A) of anthropogenic disturbance, whereas Reeves’s pheasant and Koklass pheasant were treated as subordinate species (species B) [42]. To avoid model non-convergence, it was assumed that the detection of species A and B were independent, indicating that the detection of species A at a site did not affect the probability of detecting species B at the same site, and vice versa (*r*BA = *r*Ba) [44].

The candidate models were simplified using single-species, single-season occupancy models [43]. Occupancy covariates were selected from the best models for each species and used to construct two species occupancy models. The model with the lowest AIC value was chosen as the best model from which parameters were extracted, and the species interaction factor (SIF) was calculated; SIF = 1 indicated that the spatial distribution of the two species was independent, SIF < 1 suggested spatial segregation, and SIF > 1 suggested spatial overlap [42].

#### 2.4.3. Kernel Density Estimation

Kernel density estimation was primarily used to describe the daily activity rhythms of pheasants in TM throughout the year and the impact of human disturbance on their activity rhythms [45]. To ensure the accurate timing of the behaviors, clock-recorded times were converted to solar time before the analysis [46]. The overlap coefficient was used to quantify the degree of overlap in the daily activity rhythms between species, with coefficients ranging from 0 (no overlap) to 1 (complete overlap) [47]. For sample sizes greater than 75, Δ_4_ provides the best predictive results, while for smaller sample sizes, Δ_1_ is preferred [48]. The significance level for all tests was set at *p* < 0.05.

## 3. Results

### 3.1. Environmental and Anthropogenic Associations

During model construction, all covariates exhibited collinearity that satisfied the initial assumption (Spearman’s rho < |0.7|, Table A2), thus the set of five covariates for inclusion was retained. Based on the AIC values for model selection, the optimal model for Reeves’s pheasant indicated that its site occupancy was primarily influenced by elevation, EVI, and distance to the nearest farmland. For Koklass pheasant, the optimal model showed that its site occupancy was mainly determined by elevation (ΔAIC ≤ 2, Table 2).

Different target species responded differently to environmental factors during the survey period (Table 3). Specifically, (1) the occupancy rate of Reeves’s pheasant significantly decreased as the distance to nearest farmland increased (*β* = −1.48, *p* < 0.001), indicating that the probability of the species’ occurrence decreases the farther it is from farmland, and (2) altitude had a significant positive effect on the occupancy rate of Koklass pheasant (*β* = 0.81, *p* < 0.001), meaning that the occupancy rate of Koklass pheasant increased significantly at higher altitudes.

### 3.2. Spatial Interaction

Based on the results of the two-species occupancy model (Table A3), the primary environmental factor influencing the relationship between Reeves’s pheasant and various anthropogenic disturbances was the distance to the nearest farmland. In contrast, the factors affecting the relationship of the Koklass pheasant with different anthropogenic disturbances vary, and the distribution of the Koklass pheasant with human activities and free-ranging dogs was mainly influenced by elevation, whereas its distribution with cattle was primarily affected by the interaction between elevation, distance to the nearest road, and distance to the nearest farmland.

#### 3.2.1. Spatial Interactions Between the Two Types of Pheasants

Table 3 shows that Reeves’s pheasant and Koklass pheasant tended to segregate (SIF = 0.84 ± 0.56). The probability of site use by Koklass pheasant was significantly higher when Reeves’s pheasant was present than when it was absent (*psi*BA = 0.44 ± 0.17, *psi*Ba = 0.90 ± 0.13, *p* < 0.001).

#### 3.2.2. Reflections of Two Pheasant Species on Anthropogenic Disturbance

Table 4 shows that the spatial distribution of the Koklass pheasant was independent of grazing cattle and free-ranging dogs within the reserve (SIF = 1), whereas it tended to be separate from human activities (SIF = 0.76 ± 0.79).

The spatial distribution of Reeves’s pheasant was nearly independent of human activities and grazing cattle, while it tended to overlap with free-ranging dog activity within the reserve (SIF = 1.1 ± 0.72).

### 3.3. Rhythms of Daily Activity

Diel activity analysis showed that the Reeves’s pheasant and Koklass pheasant are strictly diurnal species that are active mainly in the early morning and late afternoon (Figure 2). Under different disturbance levels, their activity peaks exhibited slight variations. In areas with low disturbance (Figure 2a), the two species had nearly overlapping peaks of activity (6:00 a.m. and 6:00 p.m.), with a low activity period around noon. The overlap between the two species during this period was 0.90 (95% CI: 0.9–0.96, *p* = 0.018).

In areas with higher human disturbance (Figure 2b), the overlap decreased slightly to 0.88 (95% CI: 0.85–0.94, *p* = 0.31). Reeves’s pheasant continued to exhibit two distinct activity peaks, with a low-activity period at noon. However, Koklass pheasant showed three peaks, with an additional small peak occurring during the midday period (12:00 p.m. to 2:00 p.m.), in addition to its early morning and late afternoon peaks.

Under high disturbance conditions (Figure 3), the overlap between Reeves’s pheasant and livestock was 0.74 (95% CI: 0.91–0.96, *p* < 0.001), with humans was 0.65 (95% CI: 0.92–0.97, *p* < 0.001), and with dogs was 0.85 (95% CI: 0.85–0.95, *p* = 0.059). For Koklass pheasant, the overlap with livestock was 0.80 (95% CI: 0.91–0.96, *p* < 0.001), with humans was 0.69 (95% CI: 0.92–0.97, *p* < 0.001), and with dogs was 0.88 (95% CI: 0.84–0.95, *p* = 0.06).

## 4. Discussion

The impact of human disturbance on wildlife in protected areas has garnered extensive attention, especially regarding free-ranging livestock within reserves [49,50,51], whereas research on the interactions between other human activities and wildlife remains relatively scarce. Our study provides insight into the coexistence of large ground-dwelling birds and the effects of human disturbances on their spatiotemporal distribution. Our findings indicate that the current levels of human activity within the reserve do not have significant negative impacts on the distribution and daily activities of Reeves’s pheasant and Koklass pheasant.

Habitat segregation is one of the most common and important forms of niche differentiation among sympatric species [52]. We found that two Galliforme species exhibited significant habitat segregation along ecological gradients. The single-species occupancy model revealed that elevation is the key factor influencing the site occupancy probability of Koklass pheasants, whereas their detection probability was not significantly affected by other environmental factors. In contrast, Reeves’s pheasants preferred low-elevation farmland areas. Results from the two-species model showed that Koklass pheasants tended to avoid sites occupied by Reeves’s pheasants (SIF < 1), supporting the hypothesis that spatial niche segregation alleviates interspecific competition [53]. Koklass pheasants’ preference for higher elevations may be driven by both intrinsic factors as well as adaptation to achieve niche segregation [5,54], reducing competition and acquiring access to more food resources. Their limited dependency on other environmental factors further highlights their adaptability to mid- and high-elevation habitats. In contrast, the habitat selection of Reeves’s pheasants appeared to be somewhat seasonal. Although previous studies suggest that they tend to avoid farmland and residential edges during the summer [22,55], we found that they preferred farmland areas during late summer (the mid-to-late breeding season). This behavior may be attributed to the abundant food resources or suitable habitat conditions provided by farmlands during this period [25], along with a reduced sensitivity to human disturbance, which facilitates rearing offspring [56].

Overall, these two pheasant species achieve significant spatial niche segregation by selecting different habitats. This segregation not only mitigates interspecific competition but also underscores the critical role of elevation gradients in shaping the vertical distribution patterns of wildlife [54,57]. The results of the species occupancy models indicate that among various human disturbances, livestock activities have the smallest impact on the two pheasant species. Their distributions remain largely independent. Previous studies have shown that omnivorous pheasants are minimally affected by livestock grazing, with their population dynamics being neutral or even positively associated with grazing intensity [58]. This is likely because of their broader dietary range and ability to utilize food resources found in livestock dung [10,59].

Because of human activity disturbance, Koklass pheasant tends to avoid sites with human presence, whereas Reeves’s pheasant is spatially independent of human activity. This may be because Koklass pheasant typically inhabits higher-altitude mountainous areas, where the environment is more fragile and more sensitive to external disturbances, leading to stronger avoidance behavior towards human interference. In contrast, Reeves’s pheasant lives in mid- to low-altitude areas with more diverse habitats, which may provide higher tolerance to disturbances [60,61]. The distribution of free-ranging dogs and Reeves’s pheasants tends to overlap. This overlap may be attributed to the Reeves’s pheasants’ habit of frequenting farmland areas, which attracts dogs within the protected area that may track their scent and pursue them. Studies suggest that predators are more active in food-rich environments and that many predators can track prey, resulting in a positive correlation between predator and prey densities [62,63]. Although the spatial distributions of dogs and Reeves’s pheasants tend to coincide, and Koklass pheasants and dogs remain spatially independent, both pheasant species may avoid areas with the highest dog activity within the dogs’ range to evade predation [52,64,65].

The diurnal activity rhythms of wildlife are influenced by various factors, including food availability, predators, reproductive demands, disturbances, weather, and temperature [66,67]. Studying these rhythms provides valuable insights into animals’ ecological behavioral strategies and interspecies relationships. In the present study, Reeves’s pheasant and Koklass pheasant were both identified as diurnal species with no nocturnal activity, exhibiting a bimodal activity pattern with peaks in the early morning and late afternoon. This pattern is commonly observed in Galliformes. Under both high- and low-disturbance conditions, Reeves’s pheasant consistently displayed a bimodal activity pattern, aligning with the findings of Hua et al. [22]. However, under low-disturbance conditions, they entered a low-activity phase earlier in the day, whereas in high-disturbance areas, the activity lull was delayed. This shift may be attributed to frequent human and dog activity during the 10:00–11:00 a.m. period in high-disturbance areas, which likely forces Reeves’s pheasants to spend more time avoiding human disturbances [68,69]. Consequently, their foraging efficiency decreases, necessitating an extended active period to acquire sufficient food resources [22,70].

In contrast, as human disturbances increased, the diurnal activity pattern of Koklass pheasants shifted from bimodal to trimodal, creating temporal separation from Reeves’s pheasants during midday. This indicates a certain level of similarity in resource utilization between the two species, leading to competition over food resources [71,72]. With increased disturbance intensity, they mitigated excessive niche overlap through temporal niche differentiation, thereby reducing interspecific competition. This phenomenon aligns with the findings of Cui et al. [73], who observed that Temminck’s tragopan (*Tragopan temminckii*) and Blood pheasant (*Ithaginis cruentus*) coexisting in the same habitat minimized interspecific competition through dietary preferences, foraging strategies, and temporal niche differentiation.

Within the protected area, Reeves’s pheasants, benefiting from their numerical advantage, likely forced Koklass pheasants to adjust their temporal niche to access sufficient food resources [72]. Moreover, many mammals adjust their foraging times to avoid human activity [74,75]. Similarly, the changes in diurnal activity patterns of Reeves’s pheasants and Koklass pheasants suggest that human activity reduced their foraging efficiency, exacerbating interspecific competition and compelling them to adapt to the environment through niche adjustments [74]. We hypothesize that the differences in the daily activity patterns of the two pheasant species within the protected area are not only related to their species-specific adaptations to human disturbance and changes in their physiological states but also to differences in their population density distributions in both space and time. These differences may also be an adaptation to varying degrees of intraspecific competition. However, this hypothesis requires verification in future studies through the acquisition of detailed population density data.

## 5. Conclusions

Our study demonstrates that human activities have differential impacts on the activity patterns and habitats of pheasants within a protected area, with varying levels of sensitivity to disturbance between two species. Reeves’s pheasant shows low sensitivity to human activities and is largely unaffected by them, whereas the Koklass pheasant exhibits a higher sensitivity, resulting in further contraction of its ecological niche. Although these alterations in activity patterns may reduce interspecific competition to some extent, they also pose challenges to population stability and recovery. Given the inevitability of habitat changes, it is crucial to enhance wildlife conservation education, promote volunteer initiatives, and raise public awareness about animal protection and ecological conservation. These efforts can help mitigate the adverse effects of human activities on the local environment and contribute to fostering more sustainable coexistence between humans and nature.

## Figures and Tables

**Figure 1 animals-15-00095-f001:**
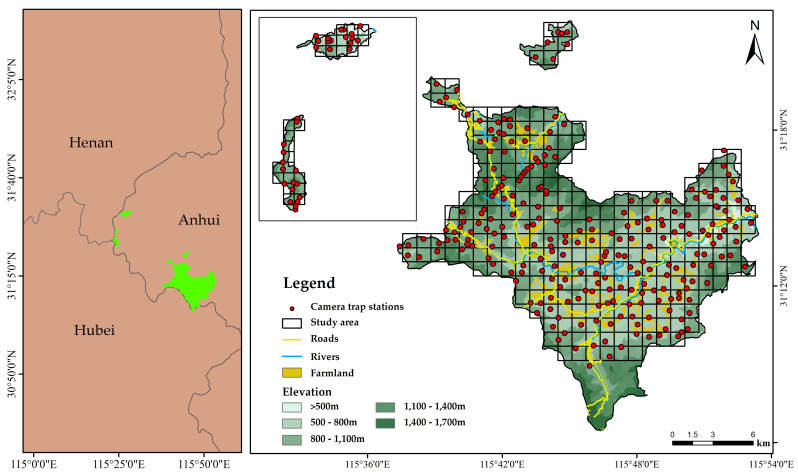
Sample area of Anhui Tianma National Nature Reserve.

**Figure 2 animals-15-00095-f002:**
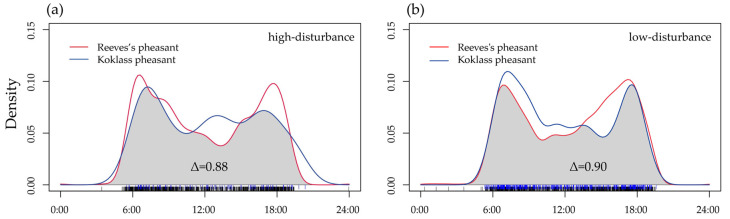
Differences in diurnal activity rhythms of the two pheasant species under different disturbance intensities. The red solid line represents Reeves’s pheasant, and the blue solid line represents the Koklass pheasant; (**a**) indicates high-disturbance areas, and (**b**) indicates low-disturbance areas. The shaded area represents the degree of overlap. Δ, overlap index.

**Figure 3 animals-15-00095-f003:**
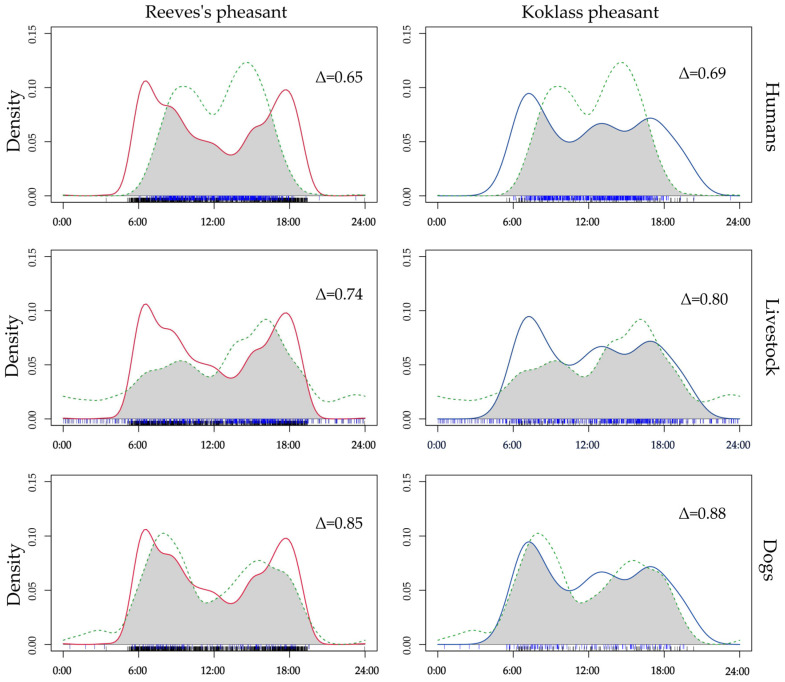
Differences in the diurnal activity rhythms between two pheasant species and human disturbance activities (including livestock, humans, and dogs). Solid curves represent the two pheasant species, while dashed curves represent human disturbances. The shaded area represents the degree of overlap. Δ, overlap index.

**Table 1 animals-15-00095-t001:** Occupancy domain model sample point covariates.

Sample covariate	Abridge	Descriptions	Source
Elevation	ELE	Infrared camera site elevation	Recorded at the time of actual deployment
Slope	SLP	Extracting Slope of Camera Locations Based on Arcgis 10.8	https://www.gscloud.cn (accessed on 24 September 2023)
Enhanced vegetation index	EVI	Landsat8 remote sensing images extracted from the Google Earth Engine website	https://earthengine.google.com (accessed on 7 October 2023)
Distance to nearest road	DTR	Extracting the distance of camera locations from the nearest road based on Arcgis 10.8	https://www.webmap.cn (accessed on 24 September 2023)
Distance to nearest farmland	DTF	Extracting the distance of camera sites from the nearest farmland based on Arcgis 10.8	https://www.webmap.cn (accessed on 24 September 2023)

**Table 2 animals-15-00095-t002:** Optimal model for occupancy based on AIC (ΔAIC≤ 2) ordering.

Model	K	AIC	ΔAIC	AIC Weight	Occupancy	Detection
Koklass pheasant						
*ψ*(ELE) *p.*	3	949.06	0	0.31	0.21	0.11
*ψ*(EVI) *p*(ELE)	4	950.53	1.47	0.15		
*ψ*(ELE) *p*(ELE)	4	950.81	1.76	0.13		
*ψ*(EVI + ELE) *p.*	4	950.84	1.76	0.13		
Reeves’s pheasant						
*ψ*(EVI + ELE) *p*(DTF)	5	3184.57	0	6.5	0.72	0.19
*ψ*(EVI + ELE) *p*(EVI + DTF)	6	3184.49	1.93	2.5		

“*ψ*”, probability of occurrence; “*p*”, probability of detection; K, number of model parameters; AIC, Akaike information criterion; ΔAIC, the relative difference in AIC values compared with the top-ranked model; AIC weight, model weight; Occupancy, occupancy rate calculated by the model; Detection, detection rate calculated by the model.

**Table 3 animals-15-00095-t003:** Effects of *β*-coefficients and standard errors (SE) based on model means on the assessment of both habitat and detectability.

Species	Model Component	Covariates	*β*	SE	*p*	
Reeves’s pheasant	Occupancy	Intercept	0.93	0.19	<0.001	***
		DTF	−1.48	0.25	<0.001	***
		EVI	0.015	0.11	0.86	
	Detection	Intercept	−1.47	0.05	<0.001	***
		EVI	0.19	0.06	0.78	
		ELE	−0.24	0.06	0.85	
Koklass pheasant	Occupancy	Intercept	−1.30	0.19	<0.001	***
		ELE	0.81	0.19	<0.001	***
		EVI	−0.01	0.08	0.86	
	Detection	Intercept	−2.05	0.11	<0.001	***
		ELE	−0.01	0.05	0.85	
		EVI	0.01	0.05	0.78	

Significance codes: *** *p* < 0.001.

**Table 4 animals-15-00095-t004:** Results of spatiotemporal interactions between dominant and subordinate species.

Species A	Species B	*psi*A	*psi*BA	*psi*Ba	SIF
Human	Reeves’s pheasant	0.53 ± 0.04	0.74 ± 0.04	0.72 ± 0.06	1.01 ± 0.77
Livestock	Reeves’s pheasant	0.32 ± 0.04	0.70 ± 0.05	0.69 ± 0.04	1 ± 1.1
Dog	Reeves’s pheasant	0.38 ± 0.04	0.84 ± 0.05	0.75 ± 0.07	1.1 ± 0.72
Reeves’s pheasant	Koklass pheasant	0.83 ± 0.04	0.44 ± 0.05	0.90 ± 0.09	0.84 ± 0.56
Human	Koklass pheasant	0.42 ± 0.05	0.21 ± 0.05	0.32 ± 0.06	0.76 ± 0.79
Livestock	Koklass pheasant	0.22 ± 0.04	0.22 ± 0.04	0.22 ± 0.04	1
Dog	Koklass pheasant	0.28 ± 0.05	0.28 ± 0.05	0.28 ± 0.05	1

*psi*A represents the occupancy rate of species A calculated by the two-species occupancy model; *psi*BA represents the occupancy rate of species B when species A is present; *psi*Ba represents the occupancy rate of species B when species A is absent; SIF represents species interaction factor.

## Data Availability

The original contributions presented in this study are included in the article. Further inquiries can be directed to the corresponding author.

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
