# Peer review of "The Spatial Relationship Between Two Sympatric Pheasant Species and Various Human Disturbance Activities"

_animals, 2025, doi:10.3390/ani15010095_

Round 1
Reviewer 1 Report
Comments and Suggestions for Authors
The manuscript serves as a research paper focusing on the spatial relationship between two sympatric pheasant species, the Reeves's pheasant and the Koklass pheasant, within the Anhui Tianma National Nature Reserve. The study offers significant insights into the dynamics between wildlife and human activities, which is essential for conservation management. With some additional details, clarifications, and a more thorough discussion of the implications, this manuscript could make a contribution to the broader field of conservation biology. The following review comments are provided to assist in further refining the manuscript. General suggestions include the following:
1. The manuscript is generally well-written, but there are occasional instances of awkward phrasing. Careful proofreading and editing for clarity and flow would be beneficial. It is suggested that the language be refined to correct any grammatical errors or spelling mistakes.
2. The methodology is well-documented, but it is suggested that a more detailed explanation be provided for the selection of covariates to better justify the model construction.
3. The research lacks quantitative analysis of the impact of changes in the intensity of human activities on the behavior of pheasants. For instance, the manuscript does not clearly differentiate the specific effects of various types and intensities of human activities on the behavior of these birds. Including such analysis would strengthen the research and its applicability.
4. Finally, the paper does not discuss how the research findings could be applied to practical nature reserve management, nor does it address specific strategies regarding how to balance the livelihoods of local residents with conservation objectives. Consider how the research findings could be applied to practical implications and discuss potential management strategies would be beneficial to explore the concrete measures or approaches that address this critical aspect of sustainable conservation practices.
Here are some specific suggestions:
Abstract
1. The abstract mentions that human activities did not significantly alter the spatial or temporal distribution patterns of the two species, but it does not specify what "varying intensities of human disturbance" were examined. It is suggested that providing more information on the types and intensities of human activities could strengthen the abstract.
Introduction
2. While the introduction mentions the importance of pheasants as indicators of ecosystem health, it could provide more context on why the specific species chosen for the study are of particular interest. It would be beneficial to include a more detailed rationale for focusing on Reeves's pheasant and Koklass pheasant specifically.
3. The introduction could briefly preview the methods used in the study, such as occupancy modeling and diel activity rhythm analysis, to give readers a sense of the approach before delving into the methods section.
4. There is a minor inconsistency in the abstract where it states that human activities did not significantly alter the distribution patterns, but the introduction suggests that human activities do have an impact. Clarifying this point would help avoid confusion.
5. Line 42-45. “Livestock farming exacerbates habitat loss and degradation[3,10,11], and altering the daily activity patterns and behaviors of wildlife[12,13].” The "and" in the sentence should connect two parallel structures, so "altering" should be changed to "alters."
Some sentence structures require adjustment for clarity, particularly in maintaining parallel verb forms within compound structures. Also, consider breaking up long paragraphs into shorter ones or reduction appropriately to improve readability.
Material and methods
6. You may want to consider reducing the use of "we" to maintain an objective tone throughout the manuscript.
7. See the former comment, it would be helpful to have more details on the criteria used for selecting the covariates in the occupancy models.
8. Figure 1: The rectangular frame around the island portion of the map of China in the upper left corner has been drawn beyond boundaries, it is necessary to revise and refine it to ensure a tidy and visually pleasing presentation. Meanwhile, adjust the legend for the elevation to be vertically aligned, and mark the unit as 'm'.
Results
9. Line 238-240. The instances where spaces are missing between the numbers and text in "Figure 3", and also before paragraphs, may affect readability. Please review the entire document to ensure proper punctuation, especially in citations and lists.
Discussion
10. The discussion effectively links your results with existing literature, but further exploration of the potential impacts on ecosystem services or biodiversity is advised. The authors could consider discussing potential limitations of the study, such as the generalizability of the results to other species or habitats, and how these might affect your results and conclusions.
11. It would be valuable to include a section discussing the potential broader ecological implications of the study's findings, such as the impact on ecosystem services or biodiversity.
Also, the manuscript could be strengthened by discussing how the findings relate to or differ from studies conducted in other regions or with different types of human disturbances.
12. Line 324-325. In this sentence, "Cui Peng" has been replaced with "Cui et al." to properly attribute the authors without specifying their full names. Please make modifications.
13. Line 329. There should be a space before the paragraph.
Conclusions
14. It would be useful to include recommendations for future research based on the study's findings.
15. Line 373. Align the image of Table A2.
References
16. Please ensure that all citations are consistent with the chosen referencing style and that there are no missing references. Please check all the references and add the doi of each reference.
Comments on the Quality of English LanguageSee the comments above.
Author Response
Responds to the reviewer’s comments:
Reviewer 1:
- The manuscript is generally well-written, but there are occasional instances of awkward phrasing. Careful proofreading and editing for clarity and flow would be beneficial. It is suggested that the language be refined to correct any grammatical errors or spelling mistakes.
REPLY: Thank you very much for your suggestions, we carefully read them in the word file. In the new manuscript, we have invited native English speakers to help us improve the manuscript. Once again, I appreciate your review and feedback.
- The methodology is well-documented, but it is suggested that a more detailed explanation be provided for the selection of covariates to better justify the model construction.
REPLY: Thank you for your valuable suggestion. In response to your feedback, we have added a more detailed explanation of the selection of covariates in lines 119–125, outlining the role of each covariate in our study and the rationale for its inclusion in the model. We believe this addition will help better demonstrate the validity of the model construction.
[Line119-L125: These covariates encompass both environmental and anthropogenic factors and comprehensively reflect the habitat characteristics and quality. (1) Elevation and slope directly influence the terrain and climatic conditions of the habitat [28]. (2) EVI indicates vegetation cover and density, which can be used to understand the ecological characteristics of the habitat. (3) Roads and human settlements are widespread disturbances in the study area and may impact habitat quality.]
- The research lacks quantitative analysis of the impact of changes in the intensity of human activities on the behavior of pheasants. For instance, the manuscript does not clearly differentiate the specific effects of various types and intensities of human activities on the behavior of these birds. Including such analysis would strengthen the research and its applicability.
REPLY: Thank you for your advice, very good advice. We will focus on strengthening this aspect in future research, particularly by distinguishing the specific effects of different types and intensities of human activity on these birds' behavior. We appreciate your suggestion, which will help us further improve our research.
- Finally, the paper does not discuss how the research findings could be applied to practical nature reserve management, nor does it address specific strategies regarding how to balance the livelihoods of local residents with conservation objectives. Consider how the research findings could be applied to practical implications and discuss potential management strategies would be beneficial to explore the concrete measures or approaches that address this critical aspect of sustainable conservation practices.
REPLY: Thanks for your suggestion. In response to your suggestion, we have emphasized the importance of strengthening wildlife conservation education for local residents to promote sustainable conservation behaviors in lines L362-368.
[L362-368: Although these alterations in activity patterns may reduce interspecific competition to some extent, they also pose challenges to population stability and recovery. Given the inevitability of habitat changes, it is crucial to enhance wildlife conservation education, promote volunteer initiatives, and raise public awareness about animal protection and ecological conservation. These efforts can help mitigate the adverse effects of human activities on the local environment and contribute to fostering more sustainable coexistence between humans and nature.]
- The abstract mentions that human activities did not significantly alter the spatial or temporal distribution patterns of the two species, but it does not specify what "varying intensities of human disturbance" were examined. It is suggested that providingmore information on the types and intensities of human activities could strengthen the abstract.
REPLY: Thank you for your valuable feedback on the abstract. In response to your suggestion, we have further clarified the information regarding "different levels of human disturbance" in the abstract and provided additional details on the types of human activities. Thank you again for your insightful recommendation. We will continue to refine the manuscript, and if you have any further comments, we would be happy to make additional improvements.
[L21-23:We used occupancy models and performed daytime activity rhythm analysis based on camera trap data to examine the spatiotemporal responses of these species to human activities, livestock, and domestic dogs.]
- While the introduction mentions the importance of pheasants as indicators of ecosystem health, it could provide more context on why the specific species chosen for the study are of particular interest. It would be beneficial to include a more detailed rationale for focusing on Reeves's pheasant and Koklass pheasant specifically.
REPLY: Thanks a lot for your reminder. We have added a more detailed explanation in lines 50–58 regarding the reasons for selecting Reeves's Pheasant and Koklass Pheasant as the study species, and why we have specifically focused on these two species. We believe they are particularly important for studying the coexistence mechanisms between human activities and wildlife, and we hope this addition helps clarify our choice of species.
[Line50-58:Reeves's pheasant (Syrmaticus reevesii) and Koklass pheasant (Pucrasia macrolopha) are large terrestrial forest birds that are under national key protection and are important components of the ecosystem [18]. Their population dynamics not only play a crucial role in maintaining ecosystem stability but also reflect the health of the local ecological environment [19]. Additionally, these two species have limited flying ability, making it difficult for them to form differentiated utilization patterns in vertical space. Thus, their ecological niches are narrow and they are sensitive to human disturbances, which may lead to intense interspecies competition [20-22]. Therefore, they are ideal groups for studying the coexistence of human activities and wildlife.]
- The introduction could briefly preview the methods used in the study, such as occupancy modeling and diel activity rhythm analysis, to give readers a sense of the approach before delving into the methods section.
REPLY: Thanks a lot. We believe the focus of the introduction should be on the research background and species selection, while the methods section will provide more detailed technical details. Therefore, we have decided not to preview the methods in the introduction. Thank you for your understanding and review.
- There is a minor inconsistency in the abstract where it states that human activities did not significantly alter the distribution patterns, but the introduction suggests that human activities do have an impact. Clarifying this point would help avoid confusion.
REPLY: Thank you very much for your reminder. Following your suggestion, we have carefully reviewed the abstract and made adjustments to the inconsistent statements to clarify this point.
[Line23-25: The results showed that human disturbance activities within the reserve impact the distribution patterns of Reeves's pheasant and Koklass pheasant, but the effect was not significant.]
- Line 42-45. “Livestock farming exacerbates habitat loss and degradation [3,10,11], and altering the daily activity patterns and behaviors of wildlife [12,13].”The "and" in the sentence should connect two parallel structures, so "altering" should be changed to "alters."
REPLY: Thanks a lot for your reminder. We have revised it following your suggestion.
[Line41-42: Livestock farming exacerbates habitat loss and degradation, and alters the daily activity patterns and behaviors of wildlife.]
- Some sentence structures require adjustment for clarity, particularly in maintaining parallel verb forms within compound structures. Also, consider breaking up long paragraphs into shorter ones orreduction appropriately to improve readability.
REPLY: Thanks, that’s a good suggestion, we have invited native English speakers to help us improve the manuscript. Once again, I appreciate your review and feedback.
- You may want to consider reducing the use of "we" to maintain an objective tone throughout the manuscript.
REPLY: Thanks, we have invited native English speakers to help us improve the manuscript. During the revision process, we will make adjustments based on your suggestions.
- Figure 1: The rectangular frame around the island portion of the map of China in the upper left corner has been drawn beyond boundaries, it is necessary to revise and refine it to ensure a tidy and visually pleasing presentation. Meanwhile, adjust the legend for the elevation to be vertically aligned, and mark the unit as 'm'.
REPLY: Thank you for your reminder. We have made the necessary revisions to the figures as per your suggestion.
- Line 238-240. The instances where spaces are missing between the numbers and text in "Figure 3", and also before paragraphs, may affect readability. Please review the entire document to ensure proper punctuation, especially in citations and lists.
REPLY: Thank you very much for your reminder. We have added the necessary spaces in the appropriate places.
- The discussion effectively links your results with existing literature, but further exploration of the potential impacts on ecosystem services or biodiversity is advised. The authors could consider discussing potential limitations of the study, such as the generalizability of the results to other species or habitats, and how these might affect your results and conclusions. It would be valuable to include a section discussing the potential broader ecological implications of the study's findings, such as the impact on ecosystem services or biodiversity. Also, the manuscript could be strengthened by discussing how the findings relate to or differ from studies conducted in other regions or with different types of human disturbances.
REPLY: Thank you very much for your suggestion. Based on your advice, we have added a discussion of the potential limitations of the study in the discussion section. We appreciate your thorough review, and if you have any further comments, we would be happy to continue making improvements.
[L350-356: We hypothesize that the differences in the daily activity patterns of the two pheasant species within the protected area is not only be related to their species-specific adaptations to human disturbance and changes in their physiological states, but also to differences in their population density distributions in both space and time. These differences may also be an adaptation to varying degrees of intraspecific competition. However, this hypothesis requires verification in future studies through the acquisition of detailed population density data.]
- Line 324-325. In this sentence, "Cui Peng" has been replaced with "Cui et al." to properly attribute the authors without specifying their full names. Please make modifications.
REPLY: Thank you very much for your reminder. We have revised it following your suggestion.
- Line 329. There should be a space before the paragraph.
REPLY: Thanks for your reminder. We have added a space before the paragraph in line 329.
- It would be useful to include recommendations for future research based on the study's findings.
REPLY: Thank you for your advice. Following your advice, I have added the limitations of this study in the discussion section and provided recommendations for future protected area development in the conclusion.
- Line 373. Align the image of Table A2.
REPLY: Thanks a lot for your reminder. We have revised it following your suggestion.
- Please ensure that all citations are consistent with the chosen referencing style and that there are no missing references. Please check all the references and add the doi of each reference.
REPLY: Thank you for your suggestion. We have added the missing DOI to the manuscript.

Reviewer 2 Report
Comments and Suggestions for Authors
This is an interesting manuscript assessing the spatio-temporal response of two pheasant species to human disturbance. A large data set is analysed and advanced statistical modelling is used to assess research questions. I have a few comments that I believe would improve this study:
L16-19. It is necessary to link nature reserves with anthropogenic disturbances.
L21 Livestock is a human activity.
L43 Remove ", and"
L119 The justification for the single-season model analysis is required at the commencement of this section.
L169 In the head of Table 3, include the full names of the covariates
L199 The results shown in the section "3.2 Spatial interaction " are not referred to in any table or figure. Are those the same results from the two-species occupancy model which are shown above?
L215 Should it be "Table 4"?
L245-253 Indicate the sample size used to fit each kernel. In the legend of Figs. 2 and 3 also explain the meaning of the delta parameter (overlap index?).
Author Response
Reviewer 2:
- L16-19. It is necessary to link nature reserves with anthropogenic disturbances.
REPLY: Thank you very much for your suggestions. We fully agree with your suggestion and recognize the importance of linking nature reserves with human disturbance. As a result, we have revised the abstract to better reflect this aspect.
- L21 Livestock is a human activity.
REPLY: Thanks a lot for your reminder. Following your suggestion, we have made the corresponding revisions in Line 20-21.
[Line20-21: Previous studies of large terrestrial birds focused primarily on livestock impacts, with less attention given to other human activities.]
- L43 Remove ", and".
REPLY: Thank you very much for your suggestions. We have made the revision as per your suggestion, changing "altering" to "alters" to ensure parallel structure in the sentence.
[Line41-42: Livestock farming exacerbates habitat loss and degradation, and alters the daily activity patterns and behaviors of wildlife.]
- L119 The justification for the single-season model analysis is required at the commencement of this section.
REPLY: Thank you for your valuable suggestion. We have reintroduced the rationale for using the occupancy model analysis at the beginning of this section to better justify its application.
[Line134-138: Infrared cameras and occupancy models are effective tools for analyzing the relationship between species distribution and habitat characteristics, particularly spatial interactions between species, niche differentiation, and the multiple factors influencing species distribution [29,30]. We used single-species occupancy models to analyze the habitat selection preferences of two pheasant species.]
- L169 In the head of Table 3, include the full names of the covariates。
REPLY: Thanks a lot for your reminder. As the full names of the covariates in this study are already fully listed in Table 1, we did not repeat them in the title of Table 3. We hope this arrangement does not cause any inconvenience.
- L199 The results shown in the section "3.2 Spatial interaction " are not referred to in any table or figure. Are those the same results from the two-species occupancy model which are shown above?
REPLY: Thank you for your suggestion. We have explained the optimal results of the two-species occupancy model in Appendix Table A3. These results are not exactly the same as those from the individual species occupancy models.
- L215 Should it be "Table 4"?
REPLY: Thank you very much for your reminder. We have made the necessary revisions.
- L245-253 Indicate the sample size used to fit each kernel. In the legend of Figs. 2 and 3 also explain the meaning of the delta parameter (overlap index?).
REPLY: Thank you for your suggestion. Since this study used data from an entire year (19 months), with a sample size far exceeding 75, we used the Δ4 model for the calculations. Additionally, regarding your comment on the meaning of the delta parameter, the "delta" parameter refers to the overlap index. We will clarify this in the figure legend to ensure that readers understand its meaning and function.

Reviewer 3 Report
Comments and Suggestions for Authors
Dear Authors,
This is an interesting and well-prepared paper, that needs clarification in methodology and results

Author Response
Reviewer 3:
- The spatial relationship between two sympatric pheasant species and various human disturbance activities.
Impact of human disturbance on spatial relationships between two pheasant species.
REPLY: Thank you for your advice. Our study primarily aims to explore how various disturbance activities affect the spatial relationships between species. The original title better reflects this focus, so we would prefer to retain the original title.
- Pheasants play an important role in maintaining ecosystem stability and their survival status can serve as a valuable indicator of the overall health of local ecosystems. However, due to their high vigilance, pheasants are difficult to locate and observe directly. In recent years, camera traps have been widely used in wildlife surveys, providing a noninvasive method that effectively visualizes species that are difficult to study.
you need to specify where?
REPLY: Thank you for your suggestion. The original wording was indeed inappropriate, and several reviewers raised similar points. We have made revisions based on their feedback. We appreciate your valuable input, which will help us further improve our research.
[Line50-58:Reeves's pheasant (Syrmaticus reevesii) and Koklass pheasant (Pucrasia macrolopha) are large terrestrial forest birds that are under national key protection and are important components of the ecosystem [18]. Their population dynamics not only play a crucial role in maintaining ecosystem stability but also reflect the health of the local ecological environment [19]. Additionally, these two species have limited flying ability, making it difficult for them to form differentiated utilization patterns in vertical space. Thus, their ecological niches are narrow and they are sensitive to human disturbances, which may lead to intense interspecies competition [20-22]. Therefore, they are ideal groups for studying the coexistence of human activities and wildlife.]
- Second, how do they respond to various human disturbances in time and space?
What were the hypotheses?
REPLY: Thanks a lot for your reminder. We have revised the original sentence to make its meaning clearer. What we intend to express is the impact of human disturbances (including humans, livestock and domestic dogs) on the two pheasant species at different times and across different spaces within the protected area.
[L76-77: how they respond to the various disturbances caused by humans, livestock, and dogs across different time periods and spatial scales.]
- We checked for collinearity among the covariates using Spearman’s rho and selected those with a correlation coefficient of |r| < 0.7 for further analysis. How it influence on results? Have You excluded any parameter?
REPLY: After calculating the correlations of the five selected covariates, the results were consistent with the initial hypothesis, with Spearman’s rho < |0.7|. Therefore, we included all the covariates in the model analysis. The correlation results for the covariates have been added in the supplementary table.
- According to model assumptions and relevant studies, livestock, humans, and dogs are treated as dominant species (species A) of anthropogenic disturbance, whereas Reeves’s pheasant and blood pheasant were treated as subordinate species (species B).
What it means?
REPLY: Thank you very much for your reminder. We have now corrected "blood pheasant" to "Koklass pheasant".
- 6. The model with the lowest AIC value was chosen as the best model from which parameters were extracted, and the species interaction factor (SIF) was calculated;
How SIF was calculated?
REPLY: The method for calculating the Species Interaction Factor (SIF) based on the occupancy rates derived from a single-season occupancy model is as follows:
In Formula, ψAB represents the shared occupancy rate when both species "A" and "B" occur simultaneously. ψA represents the occupancy rate of species "A" when species "B" is absent, and ψB represents the occupancy rate of species "B" when species "A" is absent.
- 7. To which parameter does the single "p". refer?
REPLY: "p" represents the species detection rate calculated in the occupancy model, while "p." represents the situation where, in the optimal model, the Koklass Pheasant's use of a particular site is influenced only by the occupancy rate, and is independent of the detection covariates.
- K, Number of model parameters; What it means? K is a number of parameters used for creating model? So why in model third and fourth You have the same K value?
REPLY: K represents the number of free parameters in the model, which are mainly the coefficients associated with covariates (such as elevation, EVI, etc.). Models with a higher K value may be more complex and able to explain more variability, but they also have a higher risk of overfitting. Models with a lower K value tend to be simpler.
Although models three and four have different covariates and structures, they both contain four free parameters, so their K values are the same.
- Effects of β-coefficients and standard errors (SE) based on model means on the assessment of both habitat and detectability.
Did you calculate the models with all analyzed habitat parameters?
REPLY: Yes, after calculating the correlations of the five selected covariates, the results were consistent with the initial hypothesis, with Spearman's rho < |0.7|. Therefore, we included all the covariates in the model analysis. The correlation results for the covariates have been added in the supplementary table.
[L393: Table A2 The correlation between the covariates.]
- Table 3 shows that Reeves’s pheasant and Koklass pheasant tended to segregate (0.84 + 0.56).
What that values means?
REPLY: Thank you for your reminder. This was an oversight on our part. The value represents the interspecific interaction relationship (SIF), which is an indicator used to quantify the spatial relationship between two species. It measures the way two species coexist in the same space, helping us understand their ecological interactions, such as competition, symbiosis, or other types of relationships.
[L229-231: Table 4 shows that the spatial distribution of the Koklass pheasant was independent of grazing cattle and free-ranging dogs within the reserve (SIF = 1), whereas it tended to be separate from human activities (SIF = 0.76 ± 0.79).]
- Koklass pheasants and humans tend to exhibit spatial segregation.
From this sentence, it appears that people avoid pheasants.
REPLY: Thanks a lot for your reminder. Based on your suggestion, we have revised the manuscript to further emphasize the responses of the Koklass pheasant and Reeves's pheasant to human disturbance.
[L305-309: Because of human activity disturbance, Koklass pheasant tends to avoid sites with human presence, whereas Reeves's pheasant is spatially independent of human activity. This may be because Koklass pheasant typically inhabits higher-altitude mountainous areas, where the environment is more fragile and more sensitive to external disturbances, leading to stronger avoidance behavior towards human interference.]
- Probability that the site is occupied by species A, given species A is present.
Unclear.
REPLY: Thank you for your reminder. We have carefully reviewed the relevant content and made corrections to the issues identified.
[L391: Table A1 Description of parameters in the conditional two-species occupancy model. The probability that a site is occupied by species B given the presence of species A.
Table A2 Types of human disturbance.
REPLY: Thank you very much for your reminder. We have revised it according to your suggestion.

Round 2
Reviewer 1 Report
Comments and Suggestions for Authors